# Ancient Ceramic Culture and Technological Characteristics of Xiaopi Kiln Ceramics

**Bai Mao Gong** [1] **, Khunanan Sukpasjaroen** [1] **and Thitinan Chankoson** [2,*]

1   Chakrabongse Bhuvanarth International Institute for Interdisciplinary Studies (CBIS), Rajamangala University of Technology Tawan-OK, Bangkok 20110, Thailand; gbm@zjnu.cn (B.M.G.); hokchicago@hotmail.com (K.S.)
2   Faculty of Business Administration for Society, Srinakharinwirot University, Bangkok 10110, Thailand
*   Correspondence: thitinanc@g.swu.ac.th

**Abstract:** In recent years, the Chinese government has attached great importance to the revitalization of traditional culture, and many traditional ceramic cultures have been revitalized and developed one after another. Xiaopi kiln ceramics is one of the most representative cultural symbols in Jinxi County, Jiangxi Province. Due to local economic backwardness and other reasons, the excavation of Xiaopi kiln ceramic culture has not received due attention. However, with the economic rise of Jinxi County and the people's pursuit of cultural self-confidence, the development of Xiaopi kiln ceramic culture has been supported by the local government and people. Therefore, entrusted by the Jinxi County Government, combined with the goal of unfolding the Xiaopi kiln ceramic culture, this study uses empirical research methods to carry out sampling statistics on 115 ancient ceramics unearthed using the Xiaopi kiln technique, so as to find out the technical characteristics of Xiaopi kiln ceramics, such as shape, glaze color, decorative pattern, and firing. Through descriptive analysis, this paper summarizes the industry positioning of Xiaopi kiln ceramics, which lays a theoretical foundation for the development of the Xiaopi kiln ceramic culture industry.

**Keywords:** Xiaopi kiln; ceramic modeling; decorative features; firing process





## 1. Introduction

### 1.1. Research Background

In recent years, the Chinese government has attached great importance to the development of Chinese traditional culture, especially in strengthening the inheritance and development of intangible culture. The Chinese government and its departments have successively issued many important documents and notices (Jiangxi Intangible Cultural Heritage Research and Protection Center 2021). Under the guidance of these policies, the revitalization of ancient ceramics in various parts of China has been gradually carried out, including Longquan celadon in Zhejiang Jingdezhen blue and white porcelain in Jiangxi and Jianshui purple pottery in Yunnan have begun to take shape, the number of employees has increased sharply, and the ceramic industry has gradually become their pillar industry.

Xiaopi kiln ceramics, one of the famous kilns in Chinese history, is the most representative cultural symbol in the traditional culture of Jinxi County, Jiangxi Province (Xu and Zhao 1992). It has a history of more than 2000 years. It was founded and burned in the Western Jin Dynasty, flourished in the early Northern Song Dynasty, and declined in the late Ming Dynasty. It has formed its own system of production since ancient times. It was evaluated by Lu Yu as a historical famous kiln only second to Yue kiln and Dingzhou kiln in the Tang Dynasty, ranking the third-largest famous kiln in China at that time. However, due to the relatively backward economy and the limitations of employees in Jinxi County in the early stage of China's reform and opening up, Xiaopi kiln ceramic culture has not been inherited and developed. However, with the improvement of the Jinxi County economy in recent years and the pursuit of cultural confidence of Jinxi people, Xi

Jinping, the president of China, called for the "comprehensive promotion and revitalization of Chinese traditional culture" proposed by Chinese President Xi Jinping. In July 2021, the ceramic culture heritage and development project of Xiao Po kiln was listed in the "14 five-year development plans" of the county. Jinxi County Government has signed a framework cooperation agreement with the Xiaopi kiln ceramic culture research team to jointly promote the development of Xiaopi kiln ceramic culture research, education, and tourism (Jinxi County People's Government 2021).

*1.2. Research Aim*

This research aimed to study the firing technology of traditional ceramics in the Xiaopi kiln industry, from the perspective of intangible cultural heritage, and explore the unique industry positioning of Xiaopi kiln ceramics.

*1.3. Research Scope*

First, the ceramic culture and firing process characteristics of Xiaopi kiln ceramics, including three research contents—ancient ceramic ware type, firing technology, and decorative characteristics and patterns of Xiaopi kiln ceramics—are elucidated.

Second, the study discusses the unique cultural positioning and development theory of Xiaopi kiln ceramics, so as to pave the way for research on its later development planning. Finally, The research framework of this study are presented in Figure 1.

*1.4. Research Model*

In this study, 115 ancient ceramics types, 72 ancient ceramic glaze colors, 28 ceramic decorative features and 10 firing molds of Xiaopi kiln were sampled and counted respectively. Through comprehensive analysis, the characteristics of ceramic culture and firing technology of Xiaopi kiln were obtained. The research model is shown in Figure 1.

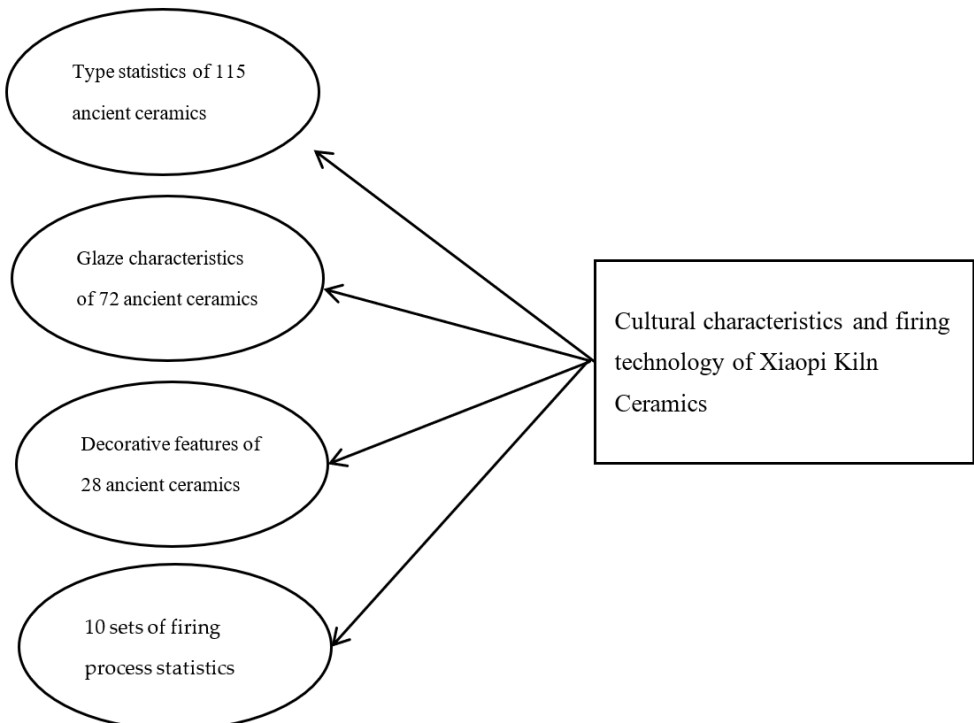

**Figure 1.** Research model.

## 2. Literature Discussion

*2.1. Overview of Xiaopi Kiln Ceramic Culture*

According to the published records of Jingdezhen ceramics (Lan 1815) and Chinese ceramics (Feng and An 2001), as well as historical documentation on Lu Jiuyuan (1139–1193),

a master of Neo Confucianism and philosophy in the Southern Song Dynasty, Xiaopi kiln ceramics are famous types of kiln ceramics in Chinese history, originated in the Wei, Jin, Southern, and Northern Dynasties and flourished in the early Northern Song Dynasty. It declined in the late Ming Dynasty and has a history of more than 2000 years. In addition, according to the records of Jinxi County annals, "Xiaopi kiln first, and then Jingdezhen" (Xu and Zhao 1992), its ancient kiln site is located in Jinxi County at the junction of today's Longhu Mountain scenic spot and the town of Duqiao. The kiln sites there spread and roll across the mountains. Standing on any mountain, one can feel the prosperity of the kiln fire in those eras. According to the above data, there are 99 folk kilns and 1 official kiln at the site, but the kiln fire stopped abruptly after only burning for more than 200 years. From the porcelain pieces, glaze colors, and firing processes excavated on-site, the kiln technology in those eras has covered today's Ding kiln, celadon, white porcelain, Ru kiln, and other characteristic processes (Zhang 2010).

*2.2. Development Status of Xiaopi Kiln Ceramic Culture*

Recently, with the improvement of the local economy and pursuit of people's cultural self-confidence, and under the call of "comprehensively carrying forward and revitalizing Chinese traditional culture" put forward by the Chinese president Xi Jinping, in July 2021, Xiaopi kiln ceramic culture inheritance and development project was listed in the "14 five-year development plans" of Jinxi County Government and became one of the key cultural projects developed by the local government. To this end, Jinxi County Government held two symposiums, attended by the main leaders of the county government, the Xiaopi kiln research team, and a number of intangible cultural heritage experts. The meeting established the framework agreement on joint cooperation between Jinxi County Government and Xiaopi kiln research team to revitalize Xiaopi kiln ceramic culture, which laid the foundation for the development of research and inheritance of Xiaopi kiln ceramic culture. However, up to now, the author has found that there is no research result on Xiaopi kiln ceramic culture in the global cultural market, and there are no systematic cultural, innovative, and practical research results. The development of the Xiaopi kiln ceramic culture industry is still in the initial stage, its development characteristics are not distinct, and the blindness of development is widespread. A series of problems such as the weak cultural heritage of employees, lack of innovation, and self-marketing initiatives have restricted the revitalization and development of Xiaopi kiln's traditional cultural industry for a long time. Therefore, research on the characteristics and positioning of Xiaopi kiln ceramics will contribute to the inheritance of Xiaopi kiln ceramic technology and industrial development planning.

## 3. Research Method

This study adopted empirical research methods. In the research process, sampling statistics and analysis were carried out on 115 pieces of Xiaopi kiln ancient porcelain pieces unearthed. The statistical types included three parts: ceramic ware type, decorative characteristics, and firing process of Xiaopi kilns. The type, glaze color, decoration, and firing techniques of these samples were set as independent variables, and the ceramic cultural characteristics and firing techniques of Xiaopi kilns were set as dependent variables. Finally, the application of statistics in social sciences was used to carry out statistics on independent variables, along with descriptive analysis, to complete the research on ceramic cultural characteristics and firing skills of Xiaopi kilns, and finally, form relevant theories.

## 4. Research Results

*4.1. Shape Characteristics of Ceramic Ware in Xiaopi Kiln*

In order to determine the characteristics of different types of Xiaopi kiln ceramics, the author sampled 115 pieces of unearthed Xiaopi kiln ancient porcelain for statistics. The sampling content included the classification of ceramic ware type and glaze color, so as to analyze the main ware type and glaze color of Xiaopi kiln ceramics. It will also summarize

the relationship between each type and glaze color in the proportion of 115 ceramics to further explain the division of ceramic types and glaze colors in Xiaopi kiln.

As can be seen from Table 1, the ceramic ware types of Xiaopi kiln sampled at this time include daily necessities such as large bowl, tea lamp, plate, small bowl, bowl, and high foot cup. This shows that in ancient times, Xiaopi kiln ceramics were mainly used for firing household appliances. Statistics also show that among the 115 samples, there are 76 large bowls, accounting for 66% of the sampling rate; 28 tea lanterns, accounting for 24.3%; 4 small bowls, accounting for 3.4%; 3 plates, accounting for 2.6%; 1 goblet, jar, holder, and small box, accounting for 0.8%. Therefore, it can be concluded that the ancient ceramics used in Xiaopi kilns are mainly living utensils such as largemouth bowls, tea lanterns, and plates.

**Table 1.** Summary types and glaze colors of ancient ceramics used in Xiaopi kilns.

| Glaze Color Vessel Shape | Big Bowl | Small Cup | Plate | Small Bowl | Goblet | Jar | Holder | Small Box | Total | Percentage% |
|---|---|---|---|---|---|---|---|---|---|---|
| Blue glaze | 31 | 7 | 1 | 1 | 1 | | | | 41 | 35.6 |
| Bluish white glaze | 41 | 12 | 2 | 3 | | | | | 57 | 49.6 |
| Mixed glaze | | 4 | | | | | | | 4 | 3.4 |
| Black glaze | 2 | 5 | | | | | | | 7 | 6.0 |
| Yellow glaze | 3 | | | | | | | | 3 | 2.6 |
| Unglazed | | | | | | 1 | 1 | 1 | 3 | 2.6 |
| total | 76 | 28 | 3 | 4 | 1 | 1 | 1 | 1 | 115 | 100 |
| Percentage% | 66.0 | 24.3 | 2.6 | 3.4 | 0.8 | 0.8 | 0.8 | 0.8 | 100 | |

In addition, in order to deeply study the modeling characteristics of the Xiaopi kiln bowls (including teacups), the author also selected 72 relatively complete instrument types from 115 samples for surveying and mapping statistics, see Table 2.

**Table 2.** Description and statistics of bowl sampling of Xiaopi kiln ceramics.

| Summary Statistics | | | | | |
|---|---|---|---|---|---|
| | n | Minimum | Maximum | Mean | Standard Deviation |
| Caliber | 72 | 3.50 | 23.10 | 13.1042 | 3.22457 |
| Bottom diameter | 72 | 2.40 | 13.30 | 5.1056 | 1.74887 |
| Height | 72 | 2.30 | 8.70 | 5.3292 | 1.33685 |
| Effective N (List status) | 72 | | | | |

From Table 2, the results show that the largest diameter of the bowl mouth is 23.10 cm, the smallest diameter is 3.5 cm, the average diameter is 13.1042 cm, and the standard deviation is 3.22457cm. The maximum bottom diameter of the corresponding bowl bottom is 13.30 cm, the minimum bottom diameter is 2.4 cm, the average bottom diameter is 5.1056 cm, and the standard deviation is 1.74887 cm. The highest height of the bowl is 8.7 cm, the lowest is 2.3 cm, the average height is 5.3292 cm, and the standard deviation is 1.33685 cm. It can be concluded that the traditional bowl type of Xiaopi kiln ceramics is mainly composed of largemouth bowls and small-bottom bowls, having characteristics of a "hat type", which is also in line with the modeling characteristics of folk bowls in the Northern Song Dynasty.

*4.2. Decorative Characteristics of Xiaopi Kiln Ceramics*

According to the above sampling results, the author selected 100 well-preserved samples of Xiaopi kiln ancient ceramics and divided the sampled Xiaopi kiln ancient ceramics into glaze decoration (72), carved decoration patterns (15), and painted decoration

patterns (13) according to different decoration types and styles. In what follows, the details are discussed.

### 4.2.1. Glaze Decorative Features

The ceramic glaze decoration of Xiaopi kiln ceramics is mainly green white glaze and green glaze, followed by yellow glaze and black glaze. The details are presented in Table 3.

**Table 3.** Description and statistics of bowl glaze color sampling of Xiaopi kiln.

| | Glaze Color Statistics of Xiaopi Kiln | | |
|---|---|---|---|
| | **Name** | **Frequency** | **Percentage** |
| | 1 | 45 | 62.5 |
| | 2 | 13 | 18.1 |
| Items | 3 | 10 | 13.9 |
| | 4 | 4 | 5.5 |
| | Total | 72 | 100.0 |

From the Table 3, the results show that among the 72 effective samples, there were 45 blue and white glazes, accounting for 62.5%; 13 green glazes, accounting for 18.1%; 10 yellow glazes, accounting for 13.9%; 4 black glazes, accounting for 5.5%. Therefore, it can be concluded that the ceramic glaze decoration of Xiaopi kiln ceramics is mainly green white glaze and green glaze, followed by yellow glaze and black glaze.

### 4.2.2. Depicting Decorative Features of Patterns

The ceramic ware decoration of Xiaopi kiln ceramics mainly focuses on depiction and painting. These two forms of decoration are common on the inner wall of the ware. The details were shown in the Table 4.

**Table 4.** Sample description statistics of Xiaopi kiln ancient ceramics depiction and decoration.

| | Depicting Decorative Patterns | | |
|---|---|---|---|
| | | **Frequency** | **Percentage** |
| | Lotus design | 5 | 33.3 |
| | Comb pattern | 5 | 33.3 |
| | Freehand cloud grass pattern | 1 | 6.7 |
| | Watergrass pattern | 1 | 6.7 |
| Items | Aquatic fish pattern | 1 | 6.7 |
| | Juban pattern | 1 | 6.7 |
| | Sunflower pattern | 1 | 6.7 |
| | Total | 15 | 100.0 |

From the Table 4, the results show that there are 7 types of patterns among 15 samples, such as lotus, comb, freehand cloud grass, and watergrass patterns. The most common patterns were lotus and comb, each with five patterns, accounting for 33.3%, respectively; the second-most-common patterns include those of freehand cloud grass, watergrass, watergrass fish, chrysanthemum petal, and sunflower, accounting for 6.66%, respectively. Therefore, we can conclude that the decorative patterns of Xiaopi kiln ceramics are mainly lotus petal patterns and comb patterns, followed by freehand cloud grass, watergrass, geometric, and so on.

### 4.2.3. Painting Pattern Decoration

From Table 5, the results show that by depicting decoration, 13 of the 100 samples of ancient ceramics are painted decoration. Among the 13 samples, 11 are painted with chrysanthemum dot patterns, accounting for 84.6%, followed by 2 geometric patterns,

accounting for 15.4%. Therefore, we can draw the following conclusion: Most of the decorative patterns are based on plant modeling. Dot decoration is generally configured next to the plant patterns, and the color is mainly blue and white oil. However, the decorative features of Xiaopi kiln ceramics include pure glaze decoration, depiction decoration, and painting decoration. Pure glaze decoration is mainly composed of green white glaze and green glaze, followed by yellow glaze and black glaze; the depiction and decoration are mainly lotus and comb patterns, followed by freehand cloud grass, watergrass, watergrass fish, chrysanthemum petal, and sunflower patterns; painting decoration is mainly chrysanthemum dot pattern decoration, followed by geometric patterns. In the collected samples, depiction decoration is displayed on the inner wall of the instrument, and the painting decoration is on the outer wall and bottom of the instrument. Both have used underglaze decoration techniques.

**Table 5.** Sample description and statistics of Xiaopi kiln ancient ceramic painting and decoration.

| | Painting Decorative Pattern | | |
|---|---|---|---|
| | | Frequency | Percentage |
| | Chrysanthemum dot pattern | 11 | 84.6 |
| Items | Geometric grain | 2 | 15.4 |
| | Total | 13 | 100.0 |

### 4.3. Firing Technology of Ceramics in Xiaopi Kilns

In order to determine the ceramic firing process of Xiaopi kilns, the author sampled 10 well-preserved firing utensils from the unearthed firing utensils used in Xiaopi kilns for statistics. Among the 10 firing utensils, there are 4 potted overlapping firing models, accounting for 40%; 2 pot-top burning models, accounting for 20%; other firing models, such as pot ring firing and pad cake firing, account for 10%, respectively. These data show that the ceramic firing process of Xiaopi kilns is mainly a pot firing process, and stack firing and overhead firing are more common in pot firing, see Table 6. This form of firing developed because ancient kilns used firewood. Potted firing can ensure uniform heating of porcelain and that the glaze will not be affected by firewood ash, which is also consistent with the firing method in the Song Dynasty.

**Table 6.** Sampling description and statistics of ancient ceramic firing process in Xiaopi kiln.

| | Statistics of Firing Appliances in Xiaopi Kiln | | |
|---|---|---|---|
| | | Frequency | Percentage |
| | Overlapping burning | 4 | 40 |
| | Upburn | 2 | 20 |
| | Branch ring burning | 1 | 10 |
| Items | Pad cake roast | 1 | 10 |
| | Sagger roast | 1 | 10 |
| | Overburning | 1 | 10 |
| | Total | 10 | 100.0 |

### 4.4. Research Summary

From the results and discussion of the above sampling statistical analysis, the ancient ceramics used in Xiaopi kilns are mainly used to make bowls, plates, tea lanterns, and cans; its glaze color is mainly celadon, celadon, and yellow glaze; the firing process is mainly characterized by pot loading, upside-down firing, stack firing, and ring firing. These distinctive features and processes cover today's Jingdezhen blue and white porcelain, Longquan celadon, Jun porcelain kiln, and other processes, which can further confirm that Xiaopi kiln ceramics indeed have high achievements in the history of Chinese ceramics.

## 5. Research Discussion

### *5.1. Discussion on Ceramic Ware Type of Xiaopi Kiln*

According to the research results discussed in Section 4, the ceramic ware types used in Xiaopi kilns are mainly living utensils such as bowls, tea lanterns, and plates. Therefore, in order to further clarify the specific process characteristics of Xiaopi kilns, the author carried out a structural analysis of these bowls and plates. It is found that the ceramic ware used in Xiaopi kilns is made by wheel drawing, and the wheel string pattern can be seen at the bottom and abdomen; the mouth edge and upper abdominal wall of the device type are thin, exerting a feeling of lightness; the lower abdominal wall and bottom are made thick, thus appearing stable and heavy. The accessories at the mouth of the bowl are made by single-mode and manual kneading; the bottom of the green body is thick and hard, and the shape is stable and dignified. The overall shape is uniform, light, and beautiful, and the shape is practical, generous, thick, and durable, as shown in Figure 2.

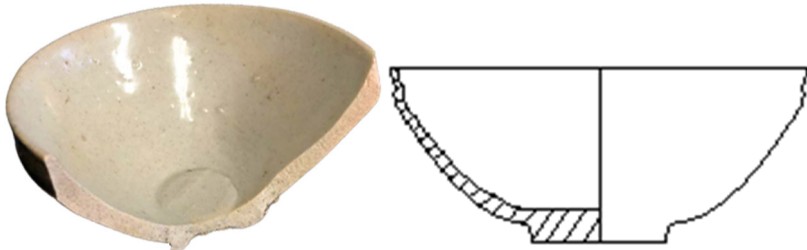

**Figure 2.** Structural anatomy of ceramics unearthed from Xiaopi kiln ceramics.

### *5.2. Discussion on Ceramic Decoration Technology of Xiaopi Kiln*

#### 5.2.1. Decorative Characteristics of Ceramic Glaze in Xiaopi Kiln

It can be concluded from the data in Figure 3.The ceramic ware types used in Xiaopi kilns are mainly green and white glaze, green glaze, and underglaze decoration. In order to refine the specific process characteristics of blue and white glaze and blue glaze, the author found that among the 72 studied samples, most of the green bodies of Xiaopi kiln ceramics are white and gray white, and the cross-section of the green body is relatively fine. It can be inferred that the porcelain clay treatment of Xiaopi kiln ceramics is relatively fine; there are small amounts of yellow glaze, black glaze, and mixed glaze in some glaze colors. The glaze has a good degree of vitrification, high brightness, and uniform glaze color. Only a small part of the glaze has characteristics of flowing glaze and accumulated glaze. This shows that the glaze application technology of ancient ceramics used in Xiaopi kilns was mainly glaze dipping, and the glaze application technology was relatively mature. In terms of glaze types, the glaze blending process of Xiaopi kiln ceramics in ancient times covers the glaze colors of Jingdezhen white porcelain and Longquan celadon today, as shown in Figure 3.

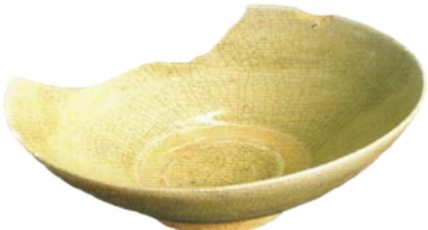

**Figure 3.** Sample of blue and white porcelain glaze unearthed from Xiaopi kiln ceramics.

#### 5.2.2. Characteristics of Ceramic Carving and Decoration Technology in Xiaopi Kiln

From the analysis results of the above 15 carved and decorated ancient porcelain, the carving and decoration of Xiaopi kiln ceramics are mainly lotus and comb patterns. Therefore, the author selected 10 pieces of ancient porcelain engraved with lotus and comb

patterns. Through surveying and mapping, it is found that the lotus pattern decoration is based on the whole of a bowl, the bottom of the bowl is taken as the lotus center, and the interior of the bowel wall is engraved with lotus petals, respectively. The lotus petals are filled with comb patterns, and the shape of each petal is different, with strong randomness. They are engraved and drawn directly on the shape by hand, as shown in Figure 4.

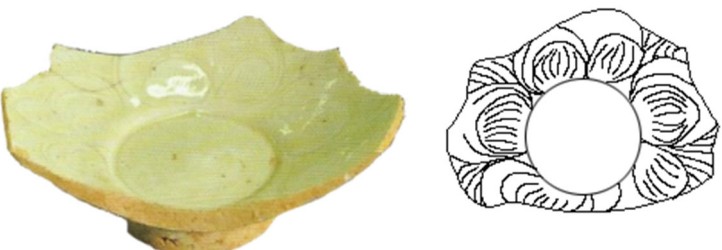

**Figure 4.** Anatomy of ceramic lotus patterns unearthed from Xiaopi kiln ceramics.

The comb pattern decoration is represented by a single group of engraving and painting, which is engraved on the inner wall of the bowl. Generally, there are three groups of comb patterns. From the patterns found, it can be seen that the maker used tools similar to a small comb to create repeated arrangements on the green body. The engraving technique is rough and random. The purpose of this decorative technique may only be to break the single glaze decorative effect, which also shows that the bowls produced by Xiaopi kiln ancient ceramics were mainly used by ordinary people, as shown in Figure 5.

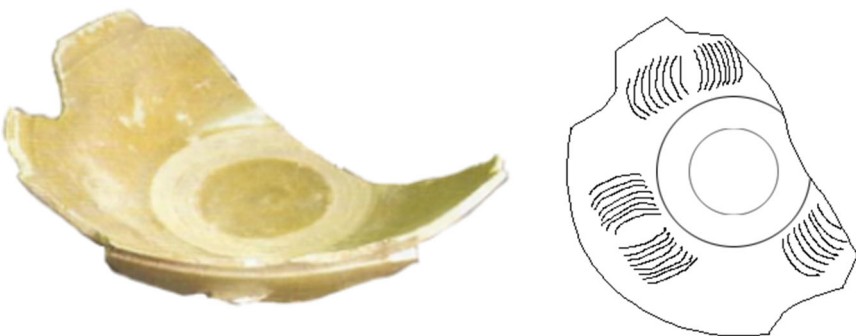

**Figure 5.** Anatomy of ceramic comb pattern samples unearthed from Xiaopi kiln ceramics.

5.2.3. Characteristics of Ceramic Painting and Decoration Technology in Xiaopi Kilns

As revealed by the 13 painted and decorated ancient porcelain samples, painting and decoration of Xiaopi kiln ceramics are mainly painted on the outer wall of the ware type and mainly chrysanthemum dot patterns. The glaze color of the chrysanthemum dot pattern is drawn with blue and white glaze, which shows that Xiaopi kiln ceramics had blue and white decoration technology in the Northern Song Dynasty. In addition, according to historical data, the blue and white porcelain in Jingdezhen first began in the Yuan Dynasty, and the blue and white porcelain in Xiaopi kiln ceramics was more than 270 years earlier than that in Jingdezhen. In addition, from the pattern of the chrysanthemum dot pattern, it can be inferred that the maker draws three freehand flowers on the outer wall of the whole bowl. Each flower is separated by a dot pattern, and the glaze color is slightly dim, as shown in Figure 6. The painting process is underglaze color painting, with simple shapes and rough and random expression methods, which reflects the characteristics of civil porcelain in ancient times.

5.3. *Discussion and Analysis of Ceramic Firing Technology in Xiaopi Kilns*

According to the 10 firing appliances sampled from Xiaopi kiln ceramics, the firing process of ancient porcelain in Xiaopi kilns is mainly pot firing, which is divided into three

processes: pad cake upside-down firing, stack firing, and combined support ring stack firing (also known as inlay firing). The characteristics of these three firing processes are further elaborated in what follows.

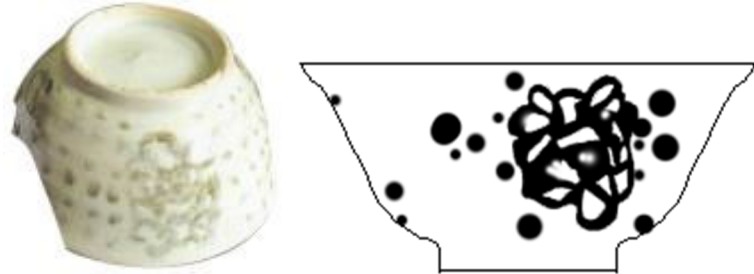

**Figure 6.** Ceramic blue and white decoration samples and drawings unearthed from Xiaopi kiln ceramics.

### 5.3.1. Baked Cake in Bowl

Cushion cake is a tool for firing pottery in Xiaopi kiln ceramics. It is a kiln tool used to isolate other pots during the firing and is often used between ceramics and pots. It is made of clay with good fire resistance, in the shape of a round cake, with a diameter greater than the bottom diameter of the placed ceramic, and the thickness is usually 3–5 cm. In order to put the ceramic on it for better stability, a circular groove is specially arranged in the middle. When in use, the bottom of the green body is placed on the pad cake, which can prevent the adhesion between the ceramic and the bowl body during firing. The pad cake is rough, reusable, and earthy yellow, which is obviously different from Gongxian kiln, Dehua kiln, and Jingdezhen kiln, because the pad cake color of Gongxian kiln in the Tang Dynasty, Dehua kiln in the Song and Yuan Dynasties, and Jingdezhen kiln in the Ming and Qing Dynasties are white, as shown in Figure 7.

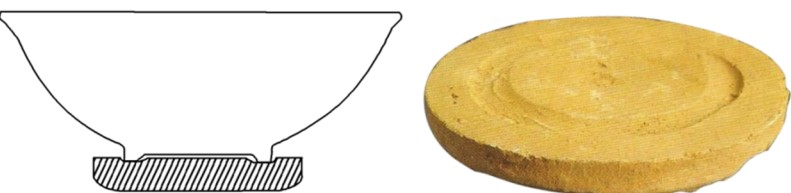

**Figure 7.** Cushion cake samples unearthed from Xiaopi kiln ceramics and drawing of upside-down firing.

### 5.3.2. Casserole Stacking

Potting and overlapping firing is a firing method in which several green bodies of the same specification are stacked together and then loaded into the bowl. The specific operation is as follows: The ring foot of the previous green body is placed at the inner bottom center of the next green body. In order to prevent green body adhesion, a circle of glaze will be scraped off inside and outside the bottom of the two green bodies. Among the 72 Xiaopi kiln ancient porcelain bowls sampled at this time, most of the bowls have more or less sand, stone, and clay at the bottom, which shows that the overlapping firing method was often used in the firing of bowls in the ancient kiln of Xiaopi kiln ceramics. This is because the output of potting and stacking firing is high. Nearly 10 bowls can be fired in one pot at a time, which indirectly shows the glory of Xiaopi kiln ceramics, as shown in Figure 8.

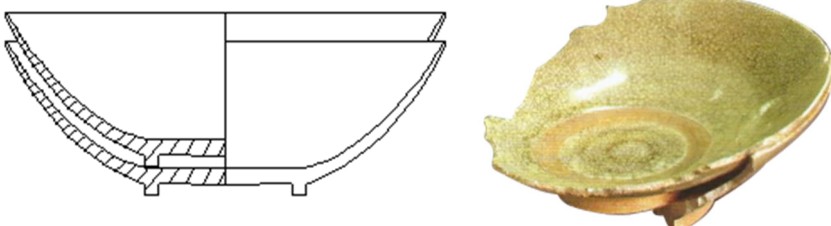

**Figure 8.** Potted and stacked firing samples unearthed from Xiaopi kiln ceramics and drawing.

### 5.3.3. Pot Loading Combined Support Ring Overburning

Support ring covering firing is a unique firing method for bowls. It makes a circular porcelain clay into a continuous section in the shape of "L". Before loading the bowl, the mouth edge of the bowl blank is sleeved on the "L" ring, and the mouth edge is not glazed. Compared with the above bowl loading and stacking firing, its advantage is that there is no contact between the bowl and the bowl, the bottom of the bowl can be glazed, the loading and firing density can be improved, the product deformation can be reduced, and the refractory materials can be saved; its disadvantage is that there is no glaze around the mouth of the bowl, but this is also a feature of the kiln in the Northern Song Dynasty. The support ring can cover 32 bowls in one turn at a time, and the firing yield is higher than that of stack firing, as shown in Figure 9.

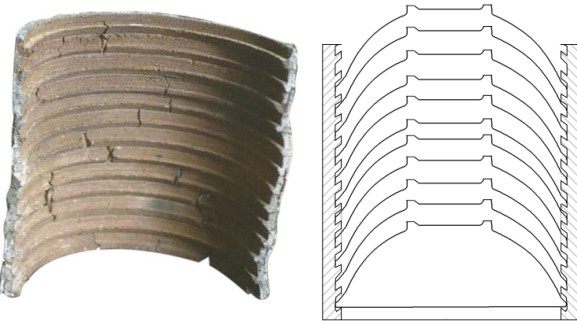

**Figure 9.** Sample and drawing of branch ring overburned excavated from Xiaopi kiln ceramics.

### 6. Research Conclusions and Recommendations

To summarize, it can be seen that there is still a large gap between the Xiaopi kiln ceramic industry and Jingdezhen white porcelain and Longquan celadon. In order to realize the rapid development of Xiaopi kiln ceramics, researchers engaged in Xiaopi kiln production and research should effectively undertake the development planning of Xiaopi kiln ceramic cultural industry based on these cultural characteristics, process characteristics, and other elements. Specific recommendations are as follows:

First, researchers should identify the inherited characteristics of Xiaopi kiln ceramic culture, integrate the local cultural, educational, and economic development needs of Jinxi County, and plan the inheritance and development of Xiaopi kiln ceramic culture. The development of Xiaopi kiln ceramic culture should first receive the strong support of the local government. The development content should also be integrated into local education, culture, and tourism activities, so as to promote local economic and cultural development and promote self-development at the same time. Another recommendation is to take advantage of cultural performances and other activities to bring Xiaopi kiln culture into the lives of the people. Finally, while inheriting the characteristics of Xiaopi Kiln Ceramics, we will diversify Xiaopi kiln ceramic products and make them enter daily life.

Second, researchers should particularly consider the application of living heritage theory and strengthen the development of cultural and creative products of Xiaopi kiln. Only by finding the cultural characteristics and technology of Xiaopi kiln ceramics and clarifying its development orientation can heritage have roots. On this basis, carry out the

living inheritance and application of Xiaopi kiln ceramic culture, and research and develop characteristic products of Xiaopi kiln that meet the needs of contemporary society. For example, promote the integration of Xiaopi kiln ceramics, electronic equipment, ecological development and other industries; Set up Xiaopi kiln ceramic experience production studio; Let Xiaopi kiln culture combine with aesthetic education into the classroom; Let Xiaopi kiln culture integrate into people's life and really "live".

**Author Contributions:** Conceptualization and methodology B.M.G., K.S. and T.C.; research design and data analysis, B.M.G., K.S. and T.C.; investigation, B.M.G. and T.C.; writing—original draft preparation, B.M.G., K.S. and T.C.; writing—review and editing, B.M.G., K.S. and T.C.; visualization and supervision, K.S. and T.C.; Correspondence, K.S. and T.C. All authors have read and agreed to the published version of the manuscript.

**Funding:** This research received no external funding.

**Institutional Review Board Statement:** Not applicable.

**Informed Consent Statement:** Not applicable.

**Data Availability Statement:** Not applicable.

**Conflicts of Interest:** The authors declare no conflict of interest.

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
