# Peer review of "Ancient Ceramic Culture and Technological Characteristics of Xiaopi Kiln Ceramics"

_arts, 2021_

Round 1

Reviewer 1 Report

This manuscript reported Chinese famous historical kilns  - "Xiaopi kiln ceramic culture and the process characteristics of the research, the foothold and research purpose, research method in empirical research on the specification, and the results of the study clearly expounds the Xiaopi kiln ancient ceramic ware, glaze color and decoration, and the characteristics of the firing technology, combined with the development of Chinese ceramic history analysis of the unique style of Xiaopi kiln, The positioning planning and theoretical support for the development of the ceramic industry of Xiaopi kiln have been done. Therefore, the research results of this paper are worth popularizing, and it is suggested to be published in your journal after minor revisions after addressing the following comments.

  1. In the article "2.2 Status quo of the development of Ceramic culture in Xiaopi kiln", the author should draw out the research problems of this paper by elaborating the status quo of the development of ceramic culture in Xiaopi kiln.
  2. It is suggested to add some research summaries at the end of the research results in Part 4 and the research discussion in Part 5, which will make the research results of the paper more clear
  3. It is suggested to simplify the research proposal and delete the last paragraph of the content of the block as the suggestion is not strongly relevant to the paper.

Author Response

Dear reviewer, Hello!

   Thank you very much for your recognition and valuable comments. I have completed the revision of the paper according to your modification requirements. Please check it! Thank you. I wish you a happy new year and good health!

  Gong baimao 

Reviewer 2 Report

An interesting account of the Ancient ceramic culture and technological characteristics of Xiaopi kiln, which will be a useful contribution to scholarship in this field. Date ranges of the ancient ceramics should be included. More engagement with other sources of recent scholarship on ICH would be advised and clearer definitions used. How has cultural inheritance transformed other traditions within ceramics or other disciplines? Who do you see as executing or overseeing the inheritance application process?  Some terms were unclear or did not translate very well, please see a list of amends below. Sustainable development is quite a big field in relation to ICH, could you provide more context around this.

Line 22. the inheritance and development of intangible culture.

Please add some footnote definitions of inheritance and the inheritors system in Chinese crafts and a definition of intangible cultural heritage. And link to this:

https://ich.unesco.org/en/what-is-intangible-heritage-00003

https://ich.unesco.org/en/RL/traditional-firing-technology-of-longquan-celadon-00205

Lines 27-28 – images of these types of pottery would be useful to see: Longquan celadon in Zhejiang Jingdezhen blue and white porcelain in Jiangxi 27 and Jianshui purple pottery in Yunnan

Lines 31 – again an image of Xiaopi Kiln Ceramics would be useful to see, is this a dragon kiln maybe title as Xiaopi (Dragon) ceramic kiln (given its name because of its resemblance to a long dragon’s body? and the red glow it creates when fired?)

Line 33 – check use of word ‘burned’? Do you mean fired?

Line 49 – can this be extended? For example – the review of a range of ancient ceramic examples and analysis of their unique characteristics etc.

Line 74. Add in a short description of this research model.

Line 87 change ‘fire’ to ‘fired’

Line 97 fix type face size

Line 97 is the cultural heritage weak or is knowledge transfer related to cultural heritage weak?

Line 98 change self-marketing thinking to self-promotion and marketing by employees?

Is employees the write term - Craftspeople?? or small independent studio makers (are they self-employed?)

101 ‘positioning are unclear change’ to ‘positioning of the Xiaopi Kiln Ceramics within the celadon culture is unclear’

104 – add in ‘blurring the boundaries of this ceramic style’

123 add in and may lead to a decline or the loss of this tradition.

140 add in date range for ancient ceramic pieces

152 and an image example of a large bowl

Line 156 change living utensil to everyday vessels such

Line 156 check word lantern?

Table 3 change caliber to calibre

Line 171 add in images of different bowls discussed

Line 174 provide date range for ancient ceramics

Line 195 provide images of different patterns used

Line 220 firing utensils – what is this? Check terms? Do you mean kiln furniture such as props/rings/saggars? – provide an image or different firing methods or add in footnote definitions.

Line 225 check term potted?

Line 238 – wheel drawing? Do you mean wheel throwing

Line 301 – check use of word civil?

Line 312 – English term might be setter or saggar - cushion cake? Unfamiliar term, consider adding in setter in brackets to add meaning. See links below:

https://www.potterycrafts.co.uk/search.php?searchterm=setter

https://en.wikipedia.org/wiki/Saggar

Line 356 check tea lanterns and cans? Cups and containers/jars?

Line 368 first inherit – add in more detail, - Firstly, this ceramic tradition should be added to the list of cultural inheritance,

Line 369 – add in plan the – change plan the future

Line 377 make them live – consider rephrasing – come alive as significant cultural artefacts

Line 384 check grammar

Line 387 add into people’s everyday lives and come alive through use

Line 389 – 390 check phrasing

Line 392 add in for future generations. Consider adding in final sentence – along the lines of it is hoped that this study has demonstrated the social and cultural significance of this ceramic tradition and the legacy this could achieve….

Author Response

Thank you very much for your support and valuable comments. I have carefully revised my paper. I have submitted the questions about grammar and English words to the foreign language editor of the journal for correction. Thank you. I wish you a happy new year and good health!

Reviewer 3 Report

The manuscript needs major revision, especially for innovative consideration academic value, related work review and reference style, and English writing.

Author Response

(The authors gave the same response as above.)
